# Development and Effect Evaluation of an Action-Oriented Interdisciplinary Weaning Protocol for Cuffed Tracheostomy Tubes in Patients with Acquired Brain Injury

**DOI:** 10.3390/healthcare12040480

**Published:** 2024-02-16

**Authors:** Katje Bjerrum, Linda-Maria Delgado Grove, Sine Secher Mortensen, Jesper Fabricius

**Affiliations:** Hammel Neurorehabilitation Centre and University Research Clinic, Department of Clinical Medicine, Aarhus University, 8450 Hammel, Denmark

**Keywords:** decannulation, weaning, protocol, brain injury, rehabilitation, interdisciplinary

## Abstract

The objective was to develop an interdisciplinary weaning protocol (IWP) for patients with tracheostomy tubes due to acquired brain injury, and to effect evaluate implementation of the IWP on decannulation rates and weaning duration. An expert panel completed a literature review in 2018 to identify essential criteria in the weaning process. Based on consensus and availability in clinical practice, criteria for guiding the weaning process were included in the protocol. Using the IWP, dysphagia is graded as either severe, moderate, or mild. The weaning process is guided through a protocol which specified the daily duration of cuff deflation until decannulation, along with recommendations for treatment and rehabilitation interventions. Data from 337 patient records (161 before and 176 after implementation) were included for effect evaluation. Decannulation rate during hospitalization was unchanged at 91% vs. 90% before and after implementation (decannulation rate at 60 days was 68% vs. 74%). After implementation, the weaning duration had decreased compared to before implementation, hazard ratio 1.309 (95%CI: 1.013; 1.693), without any increased risk of tube-reinsertion or pneumonia. Furthermore, a tendency toward decreased length of stay was seen with median 102 days (IQR: 73–138) and median 90 days (IQR: 58–119) (*p* = 0.061) before and after implementation, respectively. Scientific debate on weaning protocols for tracheostomy tubes are encouraged.

## 1. Introduction

Patients with severe acquired brain injury (ABI) may still require a cuffed tracheostomy tube once they are weaned from mechanical ventilation, due to high risk of aspiration [1,2,3,4]. Prerequisites for starting oral intake of food are reduced by having a cuffed tracheostomy tube, because the tube physiologically prevents the normal movement of the larynx, thus reducing the efficiency and safety of swallowing [5] and it reduces the patients’ ability to communicate verbally, which may cause frustration and increase the level of anxiety and risk of depression [4,6]. The tube also reduces the sensitivity of the pharynx, which may influence protection of the airways [7]. Additionally, a tracheostomy tube is associated with an increased number of medical complications, longer emergency and rehabilitation hospitalization, and increased use of health-care resources [6,8,9]. Weaning from the tracheostomy tube is therefore one of the most important rehabilitation goals for patients with severe ABI [5], and decannulation should occur as soon as possible, once the patient has obtained safe and efficient swallowing function [10].

In 2013, Warnecke et al. [11] introduced an endoscopic swallowing evaluation protocol for tracheostomy decannulation in patients with acute neurological disease. Using this protocol, the patient should fulfil all the following criteria in a stepwise manner to be eligible for decannulation: (1) No pooling or silent aspiration of saliva, (2) spontaneous swallows, (3) laryngeal sensibility/cough, (4) safe swallowing of a teaspoon of puree consistency, and (5) safe swallowing of a teaspoon of water. Such an approach may be necessary in an acute setting because a tracheostomy tube may delay hospital discharge and rehabilitation (if the rehabilitation unit is not equipped to handle tracheostomy tubes) [11]. However, in rehabilitation units which are equipped for handling patients with tracheostomy tubes, rehabilitation efforts should not wait for decannulation, but should be commenced as soon as the patient is medically stable. In such clinical settings, the decision on decannulation should not rely on assessment at a single time point, since the patients’ ability to swallow safely may fluctuate along with the patients’ awareness and fatigue level. Furthermore, the weaning process from the cuffed tracheostomy tube has been described as a multifaceted process involving complex functions and activities such as respiration, speech/phonation, swallowing, and eating. An interdisciplinary rehabilitation team is therefore recommended [5,10,12,13]. In line with this, it has been suggested that tracheostomy tube management by interdisciplinary teams result in faster and higher prevalence of decannulation [5,6,12].

Santus et al. [14] have proposed an interdisciplinary predictive decannulation score which may be applied in guiding decannulation. The score relies on clinical parameters from several disciplines, which include observations over time, e.g., the ability to tolerate tube capping for ≥24 h [14]. The study presents a detailed research-based overview of clinical criteria to consider before decannulation. However, the scoring system is hypothetical and has not been validated in clinical practice. Additionally, the scoring system is not meant to guide the weaning process but merely the decision on decannulation.

Studies have pointed out the need for a systematic yet individual tailored tracheostomy tube weaning protocol for guiding decannulation [8,12]. However, none have been described in the research literature, according to our knowledge. Based on this, the objective of this study was to develop a structured action-oriented interdisciplinary weaning protocol (IWP) which is based on scientific evidence and best practice, for patients with tracheostomy tubes due to ABI. Additionally, we aimed to evaluate the association between local implementation of the IWP and incidence and time until of decannulation from a cuffed tracheostomy tube.

## 2. Materials and Methods

### 2.1. Setting

The setting for this study is a specialized rehabilitation hospital for patients with moderate to severe ABI. At the hospital ward for early neurorehabilitation, rehabilitation starts immediately after discharge from intensive care units and once the patient is independent of mechanical ventilation. From 2017 to 2021, there were a mean of 67 patients with a cuffed tracheostomy tube admitted at the ward, yearly. Interdisciplinary teams consist of physicians, nurses, social and health-care assistants, occupational therapists, and physiotherapists, in close collaboration with speech language therapists, dieticians, and neuropsychologists.

All patients are systematically screened for dysphagia within 24 h of admission, according to hospital guidelines. Patients with a tracheostomy tube are more comprehensively assessed clinically with regards to motor and sensory functions of the face, mouth, and pharynx, encompassing dysfunction of cranial nerves 5, 7, 9, 10, 11, and 12 [15,16]. If the patient has dysphagia, a clinical assessment is performed weekly to track progression [15]. This is supplemented by a fiberoptic endoscopic evaluation of swallowing (FEES), which is performed systematically within five days of admission for all patients at the ward. Results from FEES are assessed with the Penetration Aspiration Scale (PAS) [16,17], the Yale Pharyngeal Residue Severity Rating Scale [18], and the Fiberoptic Endoscopic Dysphagia Severity Scale (FEDSS) [19].

### 2.2. Development of the Action Oriented IWP

An expert panel consisting of a physician, a nurse, two occupational therapists, and a physiotherapist from the early neurorehabilitation ward revised the hospital weaning guideline in 2018. A literature review was carried out and relevant knowledge regarding criteria of the weaning process was added to the guideline. In addition, a structured action-oriented interdisciplinary weaning protocol (IWP) to support recommendations and decisions in the weaning process was added. Due to the lack of standardized, evidence-based protocol and criteria for weaning from a tracheostomy tube [5], the expert panel assessed criteria which are prerequisites for the weaning process, along with the possibility of implementing these criteria in clinical practice.

### 2.3. Search Strategy

The research databases PubMed, Cinahl, Embase, and Scopus were searched for relevant articles dating up to February 2018, for clinical parameters related to weaning from a tracheostomy tube. The search strategy for PubMed was: search (((((((“Acquired brain injury”) OR “Brain Injuries”) OR “Brain Damage, Chronic”) OR “Brain Diseases”) OR “Cerebrovascular Disorders”) AND “tracheostomy tube”) AND “Tracheal decannulation”) OR “tracheostomy decannulation”. As the literature on tracheostomy tubes and ABI is limited, the expert panel also investigated evidence from the general population of patients with tracheostomy tubes (11).

### 2.4. Patient Population for Effect Evaluation of the IWP

The association between implementation of the action-oriented IWP and incidence of and time until decannulation was evaluated using exhaustive patient data from before and after implementation of the action-oriented IWP in a quasi-experimental study (data from January 2017 to December 2021). The inclusion criterion was a cuffed tracheostomy tube at admission for inpatient rehabilitation. Available data form medical records were age, sex, date of injury, referral diagnosis, number of days with a cuffed tracheostomy tube, reinsertion of a tracheostomy tube, pneumonia, functional independence measure (FIM), and early functional abilities (EFA). EFA is an interdisciplinary scale developed to assess basic physical and cognitive functions during rehabilitation in patients with severe acquired brain injury, in which other scales like FIM are not sensitive to change in function [20,21,22,23]. The scale comprises 20 items in four domains: vegetative, oro-facial, sensorimotor abilities, and cognitive abilities. Each item is scored on a five-point Likert scale with 1 denoting no function; 2 denoting severe disturbance; 3 denoting moderate disturbance; 4 denoting slight disturbance; and 5 denoting normal function. Thus, the sum score ranges from 20 to 100. Data on aspiration risk, communication, head control, and postural control is aggregated from EFA items (6) Swallowing, (10) Head control, (11) Postural control, and (19) Communication. Scores were dichotomized with scores 1–3 representing, e.g., no stable communication and 4–5 representing stable yes/no communication.

Differences in baseline characteristics were analyzed with exact tests for categorical variables and with *t*-tests or equivalent rank-sum tests for continuous variables. Days until decannulation and risk of pneumonia were analyzed in Cox proportional hazard models adjusted for sex. Incidence of pneumonia was defined as antibiotic treatment administered on the indication pneumonia. Length of stay was analyzed with a logarithmically transformed *t*-test. Median time in days until decannulation is presented in a two-way graph.

Data were collected from medical records as a part of quality insurance of clinical practice at the hospital. In Denmark, no ethical approval is required for studies applying solely registry-based data.

## 3. Results

### 3.1. Action-Oriented IWP

The clinical criteria included in the action-oriented IWP are presented in Table 1. The criteria should be seen as indicative, and not all criteria should necessarily be fulfilled in choosing a weaning protocol. The action-oriented IWP, which is displayed in Figure 1, covers a three-step process:A clinical assessment is carried out within 24 h and FEES within five days of admission [8,11,12,24,25,26].The interdisciplinary team [5,6,10,12,13] evaluates assessments along with data related to the protocol criteria, and categorize the patient as either having severe, moderate, or mild dysphagia [11,19].Each of the three protocols provide the interdisciplinary team with guidance on the weaning process, which must be adapted to the individual patient [8,12]. The IWP present both suggestions for cuff-deflation intervals and for treatment and therapy [10,27]. Treatment and therapy [5,28] encompass, e.g., interventions related to meal situations and oral hygiene [29,30,31], tactile stimulation [15,30,31,32], mobilization of the tongue [30,32], facilitation of swallowing [30,31], ACV [7,27], neuromuscular electrical stimulation, chin-tuck, effortful swallow, supraglottic swallow, the Mendelsohn maneuver [33], and pharmacological agents to reduce the production of saliva [5].

**Table 1 healthcare-12-00480-t001:** Protocol criteria included in the interdisciplinary weaning protocol.

Protocol Criteria	Description	Comment	References
Conscious/verbal address	Some consciousness and/or response to verbal address.	There is no consensus on whether consciousness has an impact in relation to a successful weaning from the tracheostomy tube.	[5,10,14,15]
Postural control	Able to sit upright with some degree of head control.	This is also a prerequisite for oral intake of food and liquids.	[10,15]
Saliva management	Some oral transport of saliva.	The literature indicates that some oral transport of saliva increases the chance of a successful decannulation.	[5,10,15]
Swallowing of saliva	Spontaneous or facilitated swallowing of saliva.	It has been suggested that spontaneous or facilitated swallowing of saliva has an impact on weaning from tracheostomy tubes.	[5,10,14,15]
Cough reflex and strength	Spontaneous and effective cough reflex and strength.	It is suggested that cough reflex and strength are important criteria to assess, but without having consensus on how to measure it.	[5,10,14,25,34]
Reflux/vomiting	No or little problems with reflux and vomiting.	Patients that cannot protect their lower airways are at higher risk of pneumonia if they have issues with reflux and vomiting.	[5,34]
Saliva above the cuff	Saliva above the cuff measured several times a day.	Cuffed tracheostomy tubes with a suction aid is preferred. However, there is no consensus on cutoff value on the amount of saliva above the cuff.	[35]
Respiratory frequency	<25	No obstruction of the upper respiratory tract.	[5,14,34]
Heart rate	<100	A normal resting heart rate for adults ranges 60–100 beats per minute.	[36]
Saturation	>92%	Breathing room air or with supplemented oxygen.	[5,10,34]
Infections	No active infection.	Recommended before proceeding with weaning and decannulation.	[5,37,38,39]
CO_2_ Measurement	PaCO_2_ < 60 mmHg	If deemed necessary.	[14]

Abbreviations: PaCO_2_ = partial pressure of carbon dioxide; mmHg = Millimeter of mercury.

**Figure 1 healthcare-12-00480-f001:**
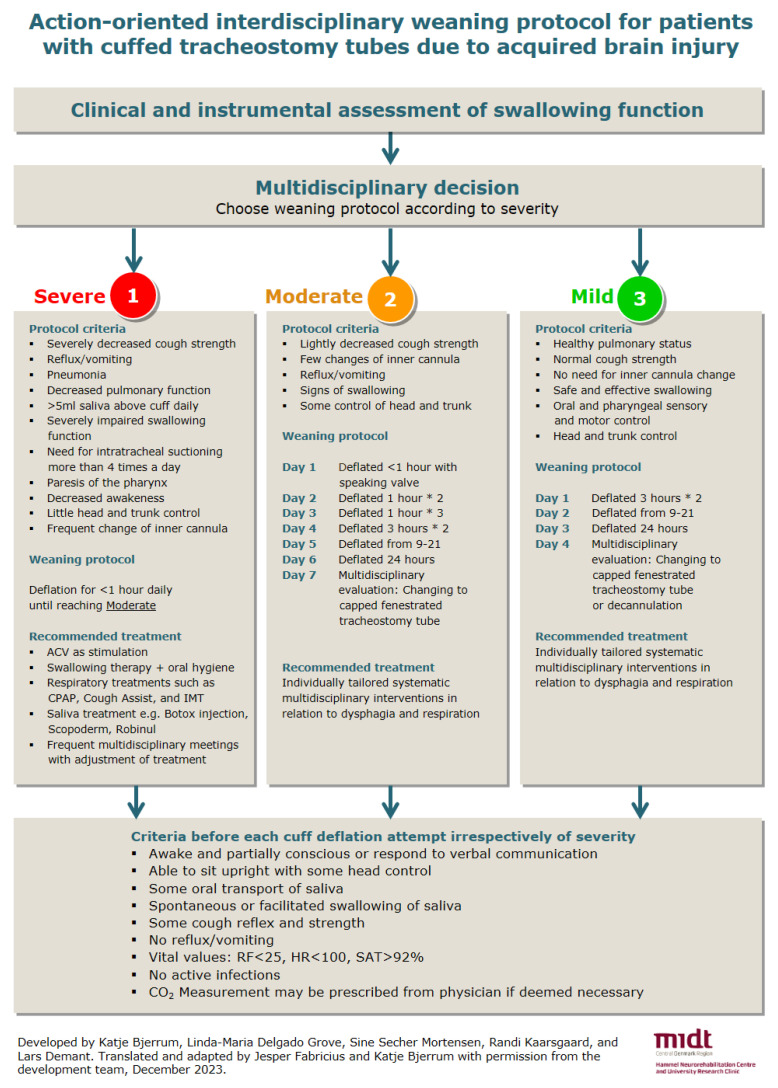
Action-oriented interdisciplinary weaning protocol for cuffed tracheostomy tubes. Comment: Criteria for the three weaning protocols are indicative and not all criteria need to be fulfilled. Severe, moderate, and mild refers to grades of dysphagia. Abbreviations: ml = milliliter, ACV = Above cuff vocalization, CPAP = Continuous positive airway pressure, IMT = Inspiratory muscle training, RF = Respiratory frequency, HR = Heart rate, SAT = Saturation. Translated and adapted by Jesper Fabricius and Katje Bjerrum with permission from ref. [40].

### 3.2. Effect Evaluation of the IWP

The action-oriented IWP was implemented in September 2019 and was introduced to all staff members at the ward [40]. The evaluation included data from a total of 337 patients, 161 before and 176 after implementation. These patients are similar in characteristics at admission (Table 2).

Decannulation rate during inpatient rehabilitation was 91% and 90% in patients before and after implementation of the IWP, respectively. Number of days from admission until decannulation from a cuffed tracheostomy tube decreased after implementation of the IWP, as illustrated in Figure 2 and analyzed in Table 3. In this analysis, subjects were right censored at discharge or at 60 days of hospitalization, whichever came first. There was a tendency toward a shorter length of hospitalization for inpatient rehabilitation, with median 102 (IQR: 73–138) and 90 (IQR: 58–119) days, before and after implementation of the IWP, respectively (*p* = 0.061). Additionally, there was a time trend showing that patients are being decannulated faster and faster in the years after implementation (Figure 3). A tracheostomy tube was reinserted in one patient before and after implementation, respectively. Risk of pneumonia was unchanged following implementation of the IWP with HR 0.899 (95%CI: 0.583; 1.385), in a Cox model adjusted for sex (Appendix A).

## 4. Discussion

Based on scientific evidence and best practice, an action-oriented IWP was developed, which proposed three separate weaning protocols based on criteria differentiating patients with severe, moderate, and mild dysphagia. The IWP was implemented in the ward for early neurorehabilitation. Data following implementation showed that patients in general were decannulated faster than before implementation of the IWP, even after adjusting for male sex, and with no increase in risk of tube-reinsertion or pneumonia. The IWP facilitates a systematic and fast-paced weaning process, which may lead to earlier decannulation and shorter hospitalization for inpatient rehabilitation. Since late 2020, the weaning process has also been facilitated by a prognostic model for decannulation which has been used at interdisciplinary conferences to discuss the prognosis of the patient [41].

Research literature indicates that an interdisciplinary team should be involved in the systematic assessment of the criteria in the weaning process [10,13] such as consciousness, oral transport of saliva, swallowing function, cough- reflex and strength [5,10,14]. In general, there are a lack of consensus and validation of these criteria, and they may therefore often rely on subjective evaluation. Consciousness is measured as drowsy or alert in the prognostic score by Santus et al., whereas the Glasgow Coma Scale (GCS) is applied as a measure of consciousness at present [42]. A study by Zanata et al. [43] showed that patients with a GCS score < 8 had insufficient protection of the airway. On the other hand, Enrichi et al. [24] do not find GCS as a crucial parameter to consider, and Steidl et al. [44] state that consciousness is not a reliable predictor of extubation. In the Facial–Oral Tract Therapy concept (F.O.T.T.^®^), wakefulness and consciousness are believed to have an impact on swallowing [10,15,32,45].

Several studies indicate that clinical swallowing assessment is important to consider before decannulation [5,14], but there is no international consensus on which clinical assessment to use. In the latest review by Medeiros et al. [46] in 2019, they found that 75% of the studies emphasized systematic swallowing assessment and 50% of the studies emphasized occlusion training, air passage permeability, and mobilization of saliva among the steps of the decannulation process.

The FEES protocol from Warnecke et al. [11] was evaluated in critically ill neurologic patients in acute care. During FEES examinations in a ward for early neurorehabilitation, saliva, spontaneous swallowing, and laryngeal sensibility/cough are taken into account. Furthermore, based on experience, successful decannulation is still possible in patients having less than one swallow per minute. Based on this, a validation of the FEES protocol proposed by Warnecke et al. in post-acute neurorehabilitation is warranted.

Several studies emphasize that effective cough is a crucial criterion for decannulation [5,14,25,34,46]. Two studies suggest measurement of peak cough flow and maximal expiratory pressure [5,14]. However, there are no recommendations on how to measure this objectively in minimally conscious patients with ABI. In a study from 2020 [47], it was suggested that cough strength should be measured with a spirometer and face-mask, which requires cuff deflation. However, a voluntary cough is not possible when the patient is minimally conscious. Reflectory cough strength can be tested by making patients inhale nebulized tussoginic agent [24]. From a global perspective, it could be advisable to reach a consensus on how cough strength is measured in minimally conscious patients with cuffed tracheostomy tubes. Future research addressing objective measures of cough strength are therefore warranted to support the interdisciplinary team with the fastest and safest weaning process.

At present, it was found that male sex was associated with longer duration until decannulation in analyses investigating the association between implementation of the IWP and time until decannulation. This is in contrast to findings from a recent systematic review, which concluded that sex was not a predictive factor for decannulation [48].

A recent study by Gallice et al. [49] have proposed a five-step multidisciplinary (pluridisciplinary) weaning protocol (MWP) for patients with ABI, which has similarities to the IWP presently proposed. The protocol includes vital stability parameters which should be considered before going from one step in the protocol to the next, which to some degree resembles those criteria which are applied before cuff deflation in the IWP. In the study by Gallice et al., a 90% decannulation rate was found, which is similar to what was found using the IWP. However, they found an impressive 7.6 days from inclusion until decannulation from a cuffed tracheostomy tube, whereas a median of 20 days was seen with the IWP, in 2021. This difference may be explained by more severely injured patients in effect evaluation of the present IWP. In the study by Gallice et al., 67% of patients were able to communicate at admission, whereas only 17% were able to stably communicate yes/no in the present study.

The study by Gallice et al. [49] took place in two neurosurgery wards. Apart from the MWP and physiotherapy, patients did not receive any rehabilitation, including swallowing therapy. Furthermore, the weaning protocol is described as multidisciplinary, which is an additive approach for incorporating knowledge from different disciplines [50]. This is a very different setting from an early rehabilitation ward in which an interdisciplinary rehabilitative approach is applied. The interdisciplinary approach analyzes, synthesizes, and harmonizes links between disciplines into a coordinated and coherent whole and can therefore been seen as an interactive rather than additive approach [50]. This is exemplified in daily practice, in which patients receive rehabilitation interventions from occupational- and physiotherapists, along with the whole interdisciplinary team having a rehabilitative approach in caring for the patient, around the clock. The IWP reflects this approach by suggesting treatment and rehabilitation initiatives for each of the protocols. It could also be argued that having three protocols which differentiate between patients with severe, moderate, and mild dysphagia creates more nuanced and individualized guidance for the weaning process.

Galice et al. [49] propose a weaning protocol which does not include an instrumental assessment of swallowing function, although they describe the benefits of a FEES in particular in diagnostics [47]. This lack of systematic assessment of swallowing using golden standard procedures and lack of swallowing therapy may explain why 35 intercurrent events in 30 subjects led to a step back in the weaning protocol. One of the criteria included in the MWP is the ability to swallow saliva (spontaneously or facilitated), in concordance with several studies [5,10,14]. Swallowing ability is essential for nutritional intake but is also directly involved in secretion management [51].

An evaluation of the MWP [49] with data from before introduction of the protocol was not presented, which makes it impossible to conclude whether introduction of the MWP improved decannulation rates and weaning duration. In evaluating the impact of the IWP, routinely gathered clinical data were collected retrospectively from medical records. This has the disadvantage that it was not possible to control which data to gather. However, it presents an objective picture of decannulation rates and weaning duration in an exhaustive patient population in a natural clinical setting, in which there have been changes in the staff managing the weaning process.

A strength of the action-oriented IWP is that it is based on current knowledge on prerequisites for a safe weaning process. Furthermore, it was developed by an expert panel of interdisciplinary health professionals with years of experience from clinical practice with ABI and cuffed tracheostomy tubes. However, a limitation of the IWP is that several criteria including cough strength have not yet been validated and objectively assessed.

## 5. Conclusions

Decannulation from a tracheostomy tube is a pivotal rehabilitation goal for patients with severe ABI. The weaning process is complex and requires interdisciplinary involvement, due to considerations on clinical parameters such as consciousness, ability to cough, swallowing function, and vital parameters. In this study, we have presented an action oriented IWP, which differentiates between patients with severe, moderate, and mild dysphagia, and includes clinical criteria for guiding the weaning process, which are based on evidence and best practice. Effect evaluation of the implementation showed that the weaning duration decreased after implementation of the IWP, without having an increase in risk of tube-reinsertion or pneumonia.

## Figures and Tables

**Figure 2 healthcare-12-00480-f002:**
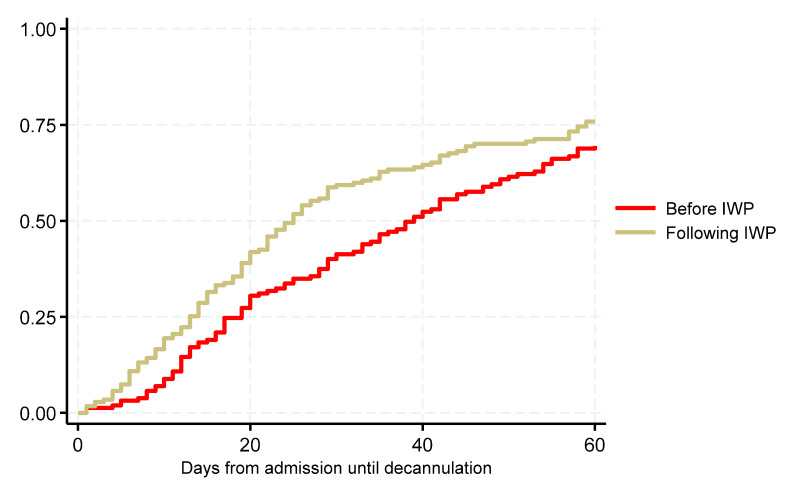
Effect evaluation of time until decannulation from a cuffed tracheostomy tube during inpatient rehabilitation before and after implementation of the action-oriented interdisciplinary weaning protocol (IWP). Right censored at discharge or 60 days of hospitalization, whichever came first.

**Figure 3 healthcare-12-00480-f003:**
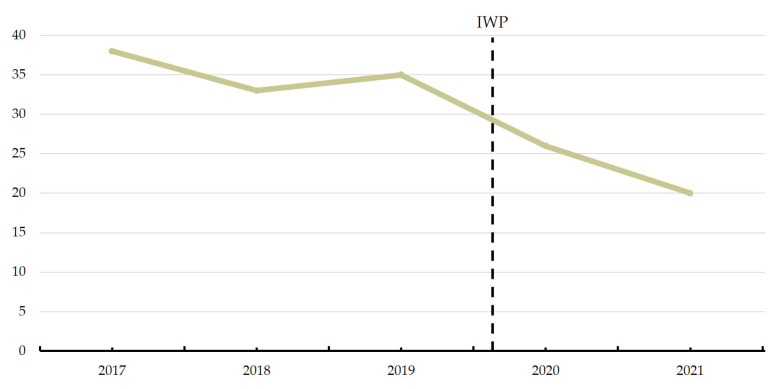
Median number of days until decannulation from a cuffed tracheostomy tube in the years before and after implementation of the action-oriented interdisciplinary weaning protocol (IWP).

**Table 2 healthcare-12-00480-t002:** Baseline characteristics of patients with tracheostomy tubes admitted before and after implementation of the action-oriented interdisciplinary weaning protocol (IWP).

	Before IWP, n = 161	After IWP, n = 176	*p*-Value
Age	56 (45–66)	55 (44–64)	0.338
Sex			0.133
• Women	29%	36%	
• Men	71%	64%	
Diagnosis			0.389
• Ischemic stroke	15%	11%	
• Hemorrhagic stroke	22%	18%	
• SAH	15%	17%	
• Stroke NOS #	4%	12%	
• TBI	27%	28%	
• Anoxic brain injury	10%	4%	
• Brain tumor	1%	3%	
• Encephalopathy NOS	7%	7%	
Day from injury until admission	31 (22–40)	31 (21–41)	0.868
FIM at admission	18 (18–21)	18 (18–20)	0.581
EFA at admission	42 (34–50)	40 (32–50)	0.324
• No aspiration risk	1%	1%	1.000
• Stable yes/no communication	20%	17%	0.571
• Head control ¤	28%	21%	0.158
• Postural control §	10%	9%	0.851

Presented as median (IQR) or percentage. Abbreviations: SAH = subarachnoid hemorrhage; TBI = traumatic brain injury: NOS = not otherwise specified; FIM = functional independence measure; EFA = early functional abilities. # Stroke NOS was excluded in the test because it is a consequence of registration practice. ¤ Able to hold head for up to 10 min. § Able to sit without support for up to 10 min.

**Table 3 healthcare-12-00480-t003:** Cox proportional hazard model on the association between implementation of the interdisciplinary weaning protocol and time until decannulation from a cuffed tracheostomy tube.

Variable	Cases/Subjects	Unadjusted HR (95%CI)	Adjusted HR (95%CI)
Weaning protocol			
• Following IWP	131/176	1.341 (1.038; 1.731)	1.309 (1.013; 1.693)
• Before IWP	111/161	Ref.	Ref.
Sex			
• Men	158/227	0.733 (0.562; 0.957)	0.753 (0.576; 0.983)
• Women	84/110	Ref.	Ref.

Abbreviations: IWP = interdisciplinary weaning protocol; HR = hazard ratio. Right censored at discharge or 60 days of hospitalization, whichever came first.

## Data Availability

Data are available within the article.

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
