# Peer review of "Development and Effect Evaluation of an Action-Oriented Interdisciplinary Weaning Protocol for Cuffed Tracheostomy Tubes in Patients with Acquired Brain Injury"

_healthcare, 2024, doi:10.3390/healthcare12040480_

Round 1

Reviewer 1 Report

Comments and Suggestions for Authors

Thank you very much for letting me review this very interesting article “Development and effect evaluation of an action-oriented interdisciplinary weaning protocol for cuffed tracheostomy tubes in patients with acquired brain injury.

The article aims to develop interdisciplinary weaning protocol for patients with tracheostomy tube post-ABI and also evaluated the effect of IWP on decannulation rate and weaning duration.

The study is interesting. The protocol is well designed and seems a promising for guiding the weaning process after ABI. Some issues however need to be addressed by the authors.

Table 2., shows the details of the patients with tracheostomy tubes admitted before and after implementation of IWP. Surprisingly, it includes only women patients, and thus the study is gender biased.

Please if the authors can discuss on this point and if they have some additional data from male patients that can be added to address the gender bias in this study. 

Author Response

Comment: Table 2., shows the details of the patients with tracheostomy tubes admitted before and after implementation of IWP. Surprisingly, it includes only women patients, and thus the study is gender biased. Please if the authors can discuss on this point and if they have some additional data from male patients that can be added to address the gender bias in this study. 

Response: Thank you for taking the time to review our manuscript. We apologize if the table caused confusion. The study included both men and women. In many journals it is common practice to only report % of one of the sexes (male/female), to have fewer rows in tables. However, we have added a row on male sex as well. From our social security number you are either categorized as male or female, and the percentage of males are therefore not surprisingly 71% and 64% in the two groups. 

Reviewer 2 Report

Comments and Suggestions for Authors

The goal of this study was to create and assess the effectiveness of an interdisciplinary IWP tailored for patients with tracheostomy tubes resulting from acquired brain injury. The IWP's impact was measured in terms of changes in decannulation rates and the duration of weaning. To form this protocol, literature review was conducted pinpointing key criteria for the weaning process. These criteria, agreed upon through consensus and practicality in clinical settings, were incorporated into the IWP. In this protocol, dysphagia is classified as severe, moderate, or mild. The weaning process is then navigated using a step-by-step approach, which includes a schedule for daily cuff deflation leading up to decannulation, coupled with specific treatment and rehabilitation strategies.

The study analyzed data from 337 patients, comparing outcomes before and after the IWP implementation (161 prior to implementation, and 176 post-implementation). The findings showed that the rate of decannulation during hospitalization remained consistent at about 90% before and after IWP implementation. However, there was an improvement in the weaning duration post-implementation, and a trend towards a reduced length of hospital stay was also noted.

Comment: the study is well-written and interesting. I just have one minor comment: did sex have an impact on the weaning (male weaning more or less than female)? How about gender? could interaction be discussed in your article?

Author Response

Comment: I just have one minor comment: did sex have an impact on the weaning (male weaning more or less than female)? How about gender? could interaction be discussed in your article?

Response: Thank you for taking the time to review our manuscript, and thank you for this comment, which we found very valuable. We have now adjusted the Cox model for sex, and there is still an effect of implementing the IWP HR 1.31 ((95%CI: 1.01;1.69), although male sex was associated with longer time until decannulation.

We have added a few lines in the methods, results, and discussion:

Methods: "Days until decannulation was presented in a Kaplan-Meier curve and analyzed in a cox proportional hazard model adjusted for sex." Line 135

Results: "Number of days from admission until decannulation from a cuffed tracheostomy tube decreased after implementation of the IWP, which was shown in a Cox proportional hazard model adjusted for sex, HR 1.31 (95%CI: 1.01;1.69)(Figure 2). See also the online appendix." Line 183-184

Discussion: "After implementation, data showed that patients in general were decannulated faster than before implementation of the action oriented IWP, even after adjusting for male sex which was associated with a longer duration until decannulation, and without having increased frequency of tracheostomy tube reinsertion." Line 206-207

Discussion: Presently, we found that male sex was associated with longer duration until decannulation in analyses investigating the association between implementation of the IWP and time until decannulation. This is in contrast to findings from a recent systematic review, which concluded that sex was not a predictive factor for decannulation [45]. Line 250-253 

Reviewer 3 Report

Comments and Suggestions for Authors

In the article titled “Development and Effect Evaluation of an Action-Oriented Interdisciplinary Weaning Protocol for Cuffed Tracheostomy Tubes in Patients with Acquired Brain Injury” authors present the development of an interdisciplinary weaning protocol (IWP) for patients with tracheostomy following acquired brain injury. The paper additionally aims to evaluate the effect of implementing the IWP decannulation rates and duration of weaning process. 

A literature review was performed by a group of experts and essential criteria in the weaning process were identified. The search resulted in forming a IWP based of a 3 step assessment - clinical examination for dysphagia within 24 hours and FEES within five days from admission, evaluation of the assessment with patients qualification for one of the severity categories ( severe, moderate or mild dysphagia) and implementation of the protocol with attention to patients individual features. Authors presented included protocol criteria in a form of table, with the comments on each criterium available. Further study assessing the results of implementation of presented IWP revealed promising results - the duration of weaning decreased after implementation of the IWP, without having more cases in which a tracheostomy tube was reinserted.

The study is novel, it summarises the available information on the tracheal tube weaning process. The proposed weaning protocol based on dysfagia severity seems clear and easy to implement in clinical environment. The article is written in comprehensive language, the structure is correct, the discussion is interesting. However it would be advised to list and address the limitations of the study. Where there any? How do the authors reflect on them? One mistake found was PaCO2 above 60 mmHg (Table 1., last line) which obviously should be below 60mmHg. 

Author Response

Comment: It would be advised to list and address the limitations of the study. Where there any? How do the authors reflect on them? One mistake found was PaCO2 above 60 mmHg (Table 1., last line) which obviously should be below 60mmHg. 

Response: Thank you for taking the time to review our manuscript, and thank you for spotting this silly mistake. We have changed this. It now says: "PaCO2 < 60 mmHg" Table 1

You are right. We have written a short paragraph of some of the strengths and limitations. Apart from having some criteria which are not objectively assessed we do not see any major limitations. The IWP has been fully implemented at the ward, and we do not see any other obvious reasons for the decline in duration until decannulation.

 "One of the strengths of the action-oriented IWP is that it is based on current knowledge on prerequisites for a safe weaning process. Furthermore, it was developed by an expert panel of interdisciplinary health professionals with years of experience from clinical practice with ABI and cuffed tracheostomy tubes. However, a limitation of the IWP, is that several criteria including cough strength are not yet validated and objectively assessed." Line 298-303

Reviewer 4 Report

Comments and Suggestions for Authors

Dear Authors,

congratulations for your paper, which would to my opinion need just few small changes.

1. please consider to shorten the first part of mat and meth / settings - and to put only relevant data in it (row 77-86)

2. please consider to put the data from row 98-105 out of  mat and meth chapter and to incorporate it in discussion chapter - if needed

3. Conclusion part should consider only of conclusions without any other comments or suggestions.

Author Response

Comment: 1. please consider to shorten the first part of mat and meth / settings - and to put only relevant data in it (row 77-86)

Response: Thank you for taking the time to review our manuscript.  The first paragraph has been shortened by three lines. We have for example deleted the sentence: "Approximately 700 patients are admitted yearly to the hospital, and the main diagnostic groups are stroke, traumatic brain injury, and encephalopathy."

Comment: 2. please consider to put the data from row 98-105 out of mat and meth chapter and to incorporate it in discussion chapter - if needed

Response: You are right. Some of this information is redundant for the reader. We have deleted this paragraph. "For 15 years, a clinical guideline for handling patients with tracheostomy tubes have been applied. However, there have not been an evidence based or structured procedure for the weaning process. The guideline was therefore revised in 2018 by an expert panel consisting of a physician, a nurse, two occupational therapists, and a physiotherapist from the early neurorehabilitation ward, along with an occupation therapist from the research unit. The members of this interdisciplinary expert panel have many years of clinical experiences in rehabilitating patients with severe dysphagia from ABI and weaning from tracheostomy tubes."

However, to not make the next paragraph too fragmented we have added this sentence at the beginning of the paragraph:

"An expert panel consisting of a physician, a nurse, two occupational therapists, and a physiotherapist from the early neurorehabilitation ward, along with an occupation therapist from the research unit, revised the hospital weaning guideline in 2018." Line 95-97

Comment: 3. Conclusion part should consider only of conclusions without any other comments or suggestions.

Response: We have deleted this paragraph: "We encourage scientific correspondence on the IWP and have made it publically available so that other institutions may use it for guiding the weaning process from a cuffed tracheostomy tube. We furthermore encourage other clinical settings managing patients with cuffed tracheostomy tube, such as the protocol proposed by Gallice et al., to share knowledge on how weaning and decannulation is guided in their clinical practice."